RESEARCH CULTURE

# Creating SPACE to evolve academic assessment

**Abstract** Universities and research institutions have to assess individuals when making decisions about hiring, promotion and tenure, but there are concerns that such assessments are overly reliant on metrics and proxy measures of research quality that overlook important factors such as academic rigor, data sharing and mentoring. These concerns have led to calls for universities and institutions to reform the methods they use to assess research and researchers. Here we present a new tool called SPACE that has been designed to help universities and institutions implement such reforms. The tool focuses on five core capabilities and can be used by universities and institutions at all stages of reform process.

**RUTH SCHMIDT, STEPHEN CURRY AND ANNA HATCH***

**\*For correspondence:**
ahatch@ascb.org

**Competing interests:** The authors declare that no competing interests exist.

## Introduction

Ghent University in Belgium made headlines in 2019 when it announced a new policy for evaluating faculty that marked a shift away from the 'rat race' of metrics and rankings towards more holistic processes focused on valuing and nurturing talent (*Redden, 2019*; *Saenen et al., 2021*). Faculty members now receive coaching from a personalized committee that evaluates them at the end of a five-year cycle. As part of the process, faculty members write narrative self-reflections to capture their significant achievements and future ambitions for research, teaching, institutional and societal engagement, and leadership.

The aim of Ghent's policy is to disrupt methods of academic assessment that are increasingly seen as an impediment to the vitality, productivity, and societal relevance of research and scholarship (*Aubert Bonn and Pinxten, 2021*). This problem has not arisen by design. Rather, it is due to a growing reliance on proxy measures of research quality in the management of recruitment, promotion, tenure and funding decisions: these proxy measures are widely used because they are convenient, not because they are meaningful. The pursuit of a higher ranking in league tables for universities has also contributed to the problem.

However, it is now widely recognized that the metric oversimplification of scholarly achievement distracts academics and institutions from broader and deeper considerations of the most important qualities of research work and culture, such as academic rigor, the rapid dissemination of results, data sharing, and mentoring the next generation of investigators (*Müller and de Rijcke, 2017*; *Hair, 2018*; *Hatch and Curry, 2020a*). Worse still, these approaches typically reward those with access to resources or insight into how to 'play the game', and it can lead institutions to prioritize rankings over their stated goals for diversity, equity, and inclusion (*Schmidt, 2020*).

Change is coming, but progress remains slow. A particular difficulty is that, despite bold initiatives in places like Ghent, point solutions and individual efforts cannot fix a flawed system. Unless a critical mass of institutions is willing to create and maintain the internal procedural and cultural conditions needed to support sustained change, efforts to define, launch, and evaluate new assessment practices are unlikely to succeed. Solving this kind of complex challenge requires a collaborative systems approach that addresses the underlying culture, infrastructure, and conditions within which assessment activities are conducted at academic institutions.

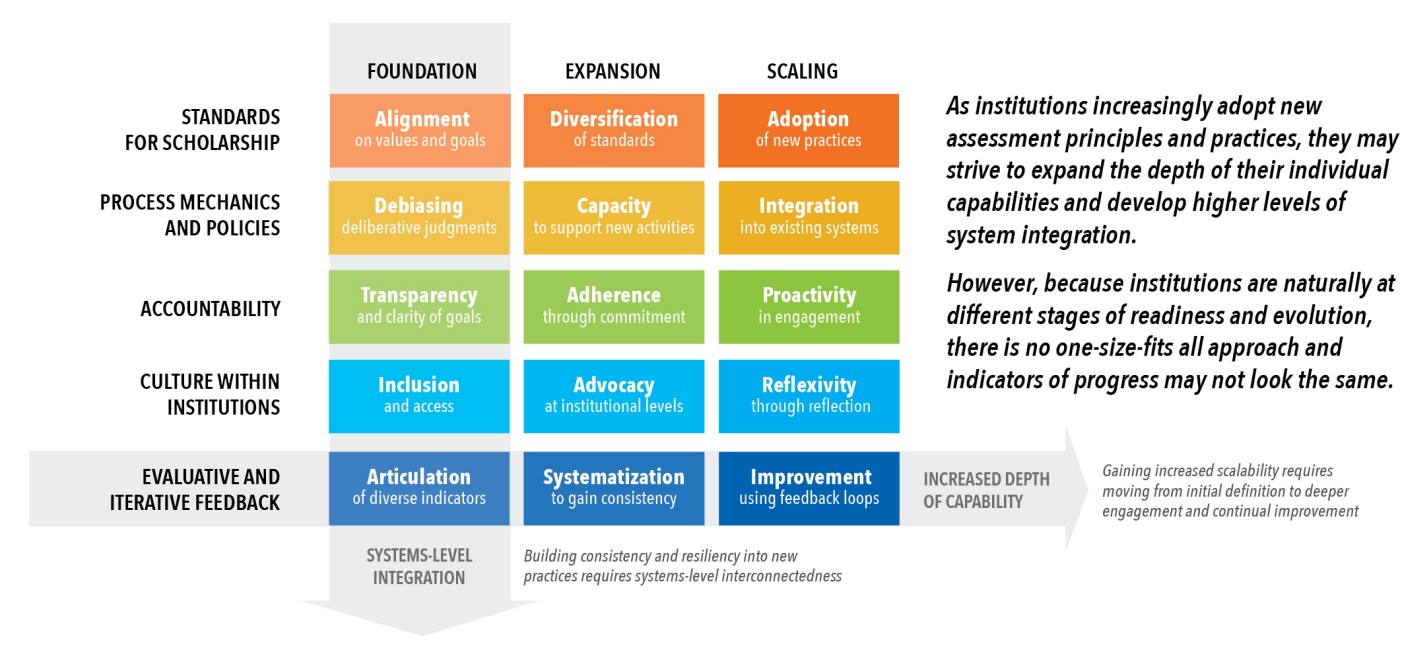

**Figure 1.** SPACE as a tool for helping universities to reform the assessment of research. SPACE is a rubric to help universities and other institutions reform how they assess research and researchers. One axis depicts five institutional capabilities that we see as critical to reforming the assessment of research: Standards for scholarship; Process mechanics and policies; Accountability; Culture within institutions; and Evaluative and iterative feedback. The other axis indicates three states of readiness for reform (foundation; basic; scaling). This figure shows an abbreviated version of the rubric; the full rubric can be seen in *Supplementary file 1*. Figure 1 is reproduced from the top panel on page 2 of *Hatch and Schmidt, 2021*.

This insight led us to develop a tool called SPACE that institutions can use to gauge and develop their ability to support new approaches to assessment that are in line with their mission and values (*Hatch and Schmidt, 2021*). SPACE can be adapted to different institutional contexts, geographies, and stages of readiness for reform, thus enabling universities to take stock of the internal constraints and capabilities that are likely to impact their capacity to reform how they assess research and researchers. SPACE builds on design principles released by DORA (an organization set up to promote best practices in the assessment of scholarly research) in 2020 (*Hatch and Schmidt, 2020b*), and a five-step approach to responsible evaluation called SCOPE that was developed by the International Network of Research Management Societies (INORMS; *Himanen and Gadd, 2019*). SPACE was developed via an iterative participatory design process that involved more than 70 individuals in 26 countries and six continents.

SPACE is a rubric that is composed of two axes (*Figure 1*). One axis depicts five core institutional capabilities that we see as critical to support more sustainable assessment practices and principles: Standards for scholarship; Process mechanics and policies; Accountability; Culture within institutions; and Evaluative and iterative feedback. The other axis indicates how ready the institution is for reform: foundation refers to institutions at the start of the process; expansion refers to institutions where the foundations are in place and the next step is to roll out reforms across institution; the third stage, scaling, refers to iteratively improving and scaling what is known to work at an institution. We envision two uses for this tool. First, it can establish a baseline assessment of the current institutional conditions, resources, and capacity to support the development and implementation of new academic assessment practices when making hiring, promotion, and tenure decisions. Secondly, the rubric can be used to retroactively analyze how the outcomes of specific interventions designed to improve academic assessment have been helped or hindered by strengths or weaknesses within the institution.

In both cases, the rubric is expressly not intended as a prescriptive mechanism, or to pass judgment on an institution's current state of academic assessment. Rather, it is designed to help institutional leaders reflect on the extent to which their organization can support sustained

and values-driven assessment practices, and where they might focus efforts to further evolve them.

To help us optimize the rubric, we piloted it with seven individuals from institutions at varying stages of reform: these individuals were selected to represent different perspectives, backgrounds and academic roles, and they included a college dean, policy advisor, research administrator, faculty member, and graduate student. This helped us identify a variety of ways and contexts in which the rubric can be used to support the development of new policies and practices. While the individuals who piloted the rubric shared valuable information about its use, they did not have the time or resources for a full cycle of implementation. The specific examples discussed in the rest of article therefore refer to the sort of outcomes we hope that the use of the SPACE rubric will lead to, not to changes made as a result of the pilot exercise.

## Early-stage reform
In piloting the rubric with individuals at institutions at the foundation stage, one positive outcome was to explicitly and systematically surface insights that had previously been suspected, but not openly shared. While this exercise may reveal or confirm unpleasant truths about the lack of readiness to make substantive change, it can also clarify potential next steps. For example, these may include aligning on values and standards of quality that should inform new assessment practices, or actively recognizing who has historically been included or excluded due to long-standing institutional norms. Although the temptation to focus on specific initiatives or try something new may be strong, simply acknowledging the need for change and using the rubric as a means to capture an honest snapshot of how things stand is a valuable first step.

We also learned that these institutions may struggle to develop the foundational capabilities needed to reform assessment practices if such reform is perceived as a mission driven by a small number of advocates who lack the seniority or resources to navigate resistance from those who are comfortable with the status quo. To enact real change, clear support from the institutional leadership is needed. But it is also important that new academic assessment processes have consistent formats and structures, to reduce any reliance on back-room channels or personal preferences in gauging 'fit'. A notable

example of this was the radical move announced by the Department of Molecular Biophysics and Biochemistry at Yale University School of Medicine in 2020: to address potential bias, candidates for tenure track Assistant Professor positions in were required to submit 'blinded' applications, which were anonymized and stripped of the names of their previous institutions, funders and the journals where they had published.

Finally, we heard consistently that it was a serious challenge to even assess the various trade-offs that would be involved in making reforms. However, the rubric can help institutions to clearly articulate and dissect perceived tensions between different institutional values or motivators, such as rankings and equity. Moreover, such discussions can help institutions better assess the trade-offs involved and identify where short-term gains may inadvertently result in missed opportunities or wasted potential in the long run. For example, the narrative CV format has shown promise as a means to recognize academic achievement and potential within under-represented groups, thereby facilitating greater equity and workforce diversity (*Lacchia, 2021*). While innovations such as narrative CVs may be embraced at a conceptual level, they can encounter resistance if they are seen as more onerous and time-consuming than existing approaches. However, by articulating clear short- and long-term goals, institutions at the early stages of reform can reinforce the value of such innovations by making clear how they are part of a broader programme.

## Mid- and late-stage reform
Institutions that have already started to reform their assessment practices can use the rubric to identify potential strengths and limitations as they seek to increase the reach and scale of these reforms. In some cases that arose during piloting, we heard that taking a critical eye to the more longitudinal effects of business-as-usual practice exposed unseen brittleness and unintended consequences. Developing new approaches to assessment reform may therefore require identifying and 'undoing' commonly accepted practices that are holding legacy systems in place. An approach to this was exemplified by the Open University in the UK, where a new promotion route that recognized and rewarded academics for public engagement was developed through iterative university-wide consultations (*Holliman et al., 2015*).

The disruption or deconstruction of current systems can also provide opportunities to reconsider whose voices are heard and valued. An example of this is the inclusion of graduate students in providing feedback during faculty searches, and even participating on search committees, as practiced by the Department of Sociology at Rutgers University. Such interventions can provide hiring committees with new kinds of insights, and they can also give early-career academics insights into aspects of academic career advancement that are normally opaque.

Given the tendency of the academic community to value research over service (*Schimanski and Alperin, 2018*), building and maintaining the capacity to instill and navigate changes to assessment practices can be a significant issue (*Saenen et al., 2021*). We heard that the work to set up and maintain new practices and principles requires dedication, time and resources. Several assessment leaders also emphasized that building capacity through upfront investment – even for relatively straightforward tasks such as aligning on values and goals – is necessary to ensure that practices are designed appropriately in the first place. This happened at the Indiana University-Purdue University Indianapolis (IUPUI) when the faculty council decided to create a new opportunity for tenure and promotion that rewards contributions to diversity, equity, and inclusion. Part of the process included the creation of a subcommittee to define clear standards of quality for work on diversity, equity, and inclusion that individual schools within IUPUI can adopt or modify to accommodate disciplinary differences (*Flaherty, 2021*).

While the ability to customize one-off solutions may be appealing, further expanding and scaling assessment efforts may require institutions at later stages of reform to develop a new capacity to balance clear standards with flexibility if they are to attract and retain a broader range of academics. For example, setting explicit expectations that potential faculty applicants should submit a minimum number of first-author papers can inadvertently dissuade otherwise qualified candidates who do not feel they fit the prescribed mold. Clear criteria are important, but may create impediments if they are not adaptable. The solution may come in landing on principles that allow individual departments or disciplines to customize their own needs while still maintaining institutional consistency. An example of this is the University of Bath, which used a 'task-and-finish' working group to develop principles of research assessment and management in 2017. These principles – that practices be contextualized, evidence-based, tailored, transparent, and centered on expert judgment – simultaneously offer mechanisms for flexibility, customization, and accountability across different disciplines.

In a similar vein, the Latin American Forum for Research Assessment (FOLEC) has articulated a set of principles to address the growing influence of Western publishing models and journal-based indicators in the humanities and social sciences. Developed following extensive community engagement led by the Latin American Council of Social Sciences (CLACSO), these principles help funders and universities in the region to balance the evergreen tension between consistency and flexibility by reconsidering and clearly defining what is meant by research 'impact'.

## Big picture

For any institution, irrespective of its readiness, the rubric will only be effective if both the positive and negative aspects of institutional conditions and infrastructure are captured. Like any assessment device, a less than honest framing of the situation will result in less useful results. Nor is the rubric a one-and-done exercise. Just as few assessment solutions or interventions work perfectly out of the gate, institutions at any stage of reform will also benefit from mechanisms for data capture, review, and improvement that are responsive to institutional shifts or the emergence of new leaders and challenges. Building in opportunities for reflexivity and reinforcement over time can help ensure that research assessment processes and cultures can adapt as necessary but also remain resilient.

How the rubric is used will depend on geographic as much as institutional context. In countries with a national classification system to assess researchers – as in Argentina and Mexico – the rubric may prove most useful for institutions in thinking about particular capabilities, such as process mechanics, accountability, and the culture within institutions. Alternatively, it could also be used at a higher level to help redefine standards for national classification systems.

In our pilot exercise, individuals from institutions at various stages of reform felt that the rubric could be used by groups of faculty and/or groups of individuals from different departments as a means to providing bottom-up input to

reform programmes being run in a top-down manner. The fact that the output of the rubric may reflect different and even contradictory perspectives depending on who is using it should be seen as a strength rather than a weakness, enabling institutions to bring the multifaceted and systemic nature of assessment activities more fully to light.

More than 20,000 individuals and organizations in 148 countries have signed the San Francisco Declaration on Research Assessment (DORA) to improve the ways research and researchers are assessed by abandoning the journal-based metrics (such as journal impact factor) for a more holistic view of academic achievement. But deciding what should be used in place of journal-based metrics, or to augment quantifiable metrics, is a more complex question that has to be solved within and across individual institutions. For example, as part of its work to facilitate collective action for responsible research assessment the Dutch Recognition and Rewards Program has brought public knowledge institutions and research funders together to align on goals.

The identification of these shared goals – such as diversifying career paths, focusing on research quality and academic leadership, and stimulating open science – has helped Dutch universities to develop broader visions for research assessment that work within their institutional contexts and capabilities. Despite this, the initiative has been met with resistance by some Dutch scholars, who are concerned the shift away from metrics like impact factors will lead to more randomness in decision-making (*Singh Chawla, 2021*). Hesitancies like these illustrate how current assessment practices need to be addressed in a constructive, evidence-based manner. By providing a framework to systematically assess and analyze how institutions are supporting the reform of research assessment practices, the SPACE rubric promises to do just that.

While the SPACE rubric was designed with academic institutions in mind, we also heard that it could be used by other organizations seeking to improve the ways decisions are made that impact research careers. For example, research funders can use the rubric to improve grant funding decisions and processes, and scholarly societies may find the rubric useful in deciding who wins awards or prizes.

We hope the SPACE rubric will encourage a wide variety of institutions to align on their values and support the development of interventions that make sense for them. We hope also that with attention, time, and input from all stakeholders, the SPACE rubric will support meaningful and persistent improvements in research assessment practices.

## Note
DORA receives financial support from eLife, and an eLife employee (Stuart King) is a member of the DORA steering committee.

## Acknowledgements
We thank everyone who informed the creation of the SPACE rubric by sharing their thoughts on fair and responsible research assessment with us and piloting various iterations of the rubric. We are especially grateful to Rinze Benedictus, Needhi Bhalla, Noémie Aubert Bonn, Nele Bracke, Elizabeth Gadd, Leslie Henderson, Miriam Kip, Andiswa Mfengu, Wim Pinxten, Olivia Rissland, Yu Sasaki, Tanja Strøm, and James Wilsdon for sharing their insights and feedback with us.

**Ruth Schmidt** is an associate professor in the Institute of Design, Illinois Institute of Technology, Chicago, United States

https://orcid.org/0000-0002-9390-8469

**Stephen Curry** is Assistant Provost (Equality, Diversity & Inclusion) and Professor of Structural Biology at Imperial College, London, UK. He is also chair of the DORA steering committee

https://orcid.org/0000-0002-0552-8870

**Anna Hatch** is the program director at DORA, Rockville, United States

ahatch@ascb.org

https://orcid.org/0000-0002-2111-3237

*Author contributions:* Ruth Schmidt, Anna Hatch, Conceptualization, Writing - original draft, Writing - review and editing; Stephen Curry, Writing - review and editing

*Competing interests:* The authors declare that no competing interests exist.

## Funding
No external funding was received for this work.

## Decision letter and Author response
Decision letter https://doi.org/10.7554/eLife.70929.sa1
Author response https://doi.org/10.7554/eLife.70929.sa2

## Additional files

### Supplementary files

• Supplementary file 1. The SPACE rubric SPACE is a rubric for analyzing institutional progress indicators and conditions for success. It was designed for universities and other institutions who want to reform how they assess research and researchers.

### Data availability

The rubric is the result of a participatory design process, rather than resulting from a qualitative research study. There are no associated datasets with the work.

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
