## [Decision Letter]

Thank you for submitting your article "Creating SPACE to evolve academic assessment" to *eLife* for consideration as a Feature Article. Your article has been reviewed by two peer reviewers, and their comments are below. The following individual involved in review of your submission has agreed to reveal their identity: Fernanda Beigel (Reviewer #2).

I would like to invite you to submit a revised manuscript that address the points raised by the reviewers (please see points a-e), and also a number of editorial points (points f-i).*Reviewer #1:*

The paper outlines a comprehensive tool (rubric) to support research performing organisations to enable and embed culture change related to academic assessment, hiring, promotion and tenure. The purpose of the tool is to shine a light on organisational conditions which can support, or hinder, initiatives to achieve culture change. The tool can be used by organisations who are in any stage of academic assessment reform, and the paper includes some guidance to support users.

The paper sets the scene very effectively, and clearly raises the issues around research assessment criteria and processes used for academic assessment. It provides a good rationale why a research organisation should take a systemic approach, noting that the conditions across an organisation are key to achieving scalable culture change.

The SPACE rubric has been developed through extensive consultation, with input from many organisations across various stages of academic assessment reform. The method used to develop the rubric is robust and reflects the needs of research organisations.

The tool has an important role to play to support research organisations to understand how the environment can hinder and support change. The rubric is presented for senior leaders to reflect on how their organisations can support values-driven assessment practices and how to evolve these. However, throughout the pilot process organisations saw potential for this to be used by faculties or groups rather than as a top-down enquiry. Further, the tool could be used by other stakeholders who either deliver assessments of academics, or who have an influence on research organisation practices. More could be said about both of these potential uses (please see points a and b below).

This tool has the potential to surface the barriers which are faced by institutions across their systems. This information will be of great benefit to organisations in their decision making, prioritisation, and goal setting.*Reviewer #2:*

This paper is a proposal made by DORA aiming to help institutions develop their ability to support new assessment policies and practices in line with their missions and values. It is a tool, called SPACE to evolve academic assessment, that builds on the principles released by DORA in 2020, which provide a starting point to help institutions experiment with and develop better practices.

This is a well informed piece that starts by acknowledging that the change proposed by DORA Declaration, shared by a many professors and institutions around the world, is evolving, but progress remains slow. Unless a critical mass of institutions are willing to create and maintain the internal procedural and cultural conditions needed to support sustained change are unlikely to succeed. It advocates for boosting efforts to improve the ways research and researchers are assessed and recommends abandoning the journal-based metrics for a more holistic view of academic achievement.

According with the authors, SPACE is a structure that aims explicitly to support the development of responsible research assessment practices in different countries/institutions. Even though this tool is the result of a survey made with 75 individuals in 26 countries, I believe it would be interesting if the authors stress more on the contextualization process that is needed in order for this tool to be used across a wide range of institutional contexts and geographies (please see points d and e below).

Points to be addressed

a) Please say more about how the rubric could be used in both a top-down and bottom-up manner at the same university or institution.

b) Please consider saying something about how the tool could be used by other stakeholders who either deliver assessments of academics, or who have an influence on research organisation practices.

c) It is not clear how the examples given in the sections "Early-stage reform" and "Mid- and late-stage reform" (eg, the examples from Yale, Open University, Rutgers, IUPUI, Bath, FOLEC/CLACSO) relate to the SPACE rubric. Please be clear about which of these examples are taken from the pilot, and which are examples of the sort of changes that you hope that use of the SPACE rubric will lead to.

d) Some countries [such as Argentina and Mexico] have a national classification system for researchers, whereas others only assess researchers at an institutional level: please comment on what this could imply for the rubric?

e) It would also be of interest if the authors can expand a bit on the experiences mentioned in piloting the rubric. How many individuals were involved in the piloting? Was there any institution as a whole involved?

f) Please explain/clarify how the language of early-stage/mid-stage/late-stage used in the text relates to the language of foundation/expansion/scaling in the rubric.

g) Lines 199-112: Please be more explicit about how SPACE could be deployed to resolve tensions (like the tensions between rankings and equity).

h) Lines 181-187: Please consider mentioning that there has been some pushback against the Recognition and Rewards Program in the Netherlands.

i) Please think about the widespread use of the word "researcher" in the article given that the article is arguing that individuals should be assessed on activities other than research.

---

## [Author Response]

Points to be addresseda) Please say more about how the rubric could be used in both a top-down and bottom-up manner at the same university or institution.

We added more information to clarify how the SPACE rubric can support top-down and bottom-up change at the same institution.

b) Please consider saying something about how the tool could be used by other stakeholders who either deliver assessments of academics, or who have an influence on research organisation practices.

We added a sentence offering ideas we heard of how the SPACE rubric might be of use to research funders and scholarly societies.

c) It is not clear how the examples given in the sections "Early-stage reform" and "Mid- and late-stage reform" (eg, the examples from Yale, Open University, Rutgers, IUPUI, Bath, FOLEC/CLACSO) relate to the SPACE rubric. Please be clear about which of these examples are taken from the pilot, and which are examples of the sort of changes that you hope that use of the SPACE rubric will lead to.

The examples listed are *not* taken from the pilot, but are the types of outcomes we hope use of the SPACE rubric will lead to. We added a sentence to clarify this point.

d) Some countries [such as Argentina and Mexico] have a national classification system for researchers, whereas others only assess researchers at an institutional level: please comment on what this could imply for the rubric?

Specific use of the rubric will depend on regional as much as institutional context. We added a sentence to clarify this point and expanded on how the rubric might be approached in areas with national classifications systems.

e) It would also be of interest if the authors can expand a bit on the experiences mentioned in piloting the rubric. How many individuals were involved in the piloting? Was there any institution as a whole involved?

Seven individuals from different institutions piloted the rubric for us. No institution as a whole piloted the rubric. We added a paragraph to describe the piloting process in more detail.

f) Please explain/clarify how the language of early-stage/mid-stage/late-stage used in the text relates to the language of foundation/expansion/scaling in the rubric.

Institutions are at different stages of readiness for research assessment reform and therefore may benefit from different types of activities to improve policy and practice. Institutions at the early stages of reform are more likely to benefit from developing foundational capabilities, whereas focusing on building expansion and scaling capabilities are more likely to be helpful for institutions at the mid- and late-stages of reform. We clarified this point in the text.

g) Lines 199-112: Please be more explicit about how SPACE could be deployed to resolve tensions (like the tensions between rankings and equity).

We updated the text to more clearly articulate how the SPACE rubric can be used to resolve tensions, such as university rankings and equity.

h) Lines 181-187: Please consider mentioning that there has been some pushback against the Recognition and Rewards Program in the Netherlands.

We added a few sentences recognizing and expanding on the resistance to the Recognition and Rewards Program in the Netherlands.

i) Please think about the widespread use of the word "researcher" in the article given that the article is arguing that individuals should be assessed on activities other than research.

We agree the use of the term “researcher” can be misleading. Throughout the text we replaced “researcher” with “academic” or “scholar.” There were a few instances where we left “researcher,” because it felt more accurate given the context of the sentence.